# Adaptive Robust Control for Quadrotors with Unknown Time-Varying Delays and Uncertainties in Dynamics

**Viswa Narayanan Sankaranarayanan *** , **Sumeet Satpute** and **George Nikolakopoulos**

Robotics and AI Team, Department of Computer, Electrical and Space Engineering,
Luleå University of Technology, 971 87 Luleå, Sweden
* Correspondence: vissan@ltu.se

**Abstract:** This article proposes an adaptive controller for a quadrotor UAV for carrying unknown payloads while tracking any trajectory. The proposed adaptive controller is robust to modeling uncertainties and does not require any a priori knowledge of the bounds of the uncertainties. The controller is also robust to time-varying delays without any constraint on the derivative of the time delay. In addition, the stability of the closed-loop system is analyzed via a Lyapunov-like method. The controller's performance is verified using a simulated quadrotor model in MATLAB in three different scenarios with varying time delays and parametric uncertainties.

**Keywords:** autonomous robots; quadrotor uav; adaptive robust controller; time-delay; nonlinear control





## 1. Introduction

Quadrotors are potential assistance in autonomous exploration, surveillance, manipulation and disaster mitigation tasks (cf. [1–7]). Due to the shifting trend in the inspection and maintenance industries, the demand for autonomous drones in confined spaces is growing rapidly. Deploying quadrotors in tasks such as inspecting tunnels, pipelines, windmills, and transmission lines, reduces the workload, and ensures human safety in hazardous environments. The overall control schemes for these quadrotors are critical, mainly due to the complexity of the operations and the types of equipment they carry that create a varying payload.

In many of these applications, the drone has to operate in a remote environment with no access to GPS or other convenient localization systems. In such scenarios, the quadrotor has to rely on visual–inertial sensors for positioning and navigation. Despite the recent advancements in the navigation industry to reduce the computational complexities, such as [8], the out-growing inspection and manipulation applications are demanding intensive processing. Such heavy processing requirements increase the drone's mass and consume much power, which is undesirable for operations involving longer flight time. Moreover, since technologies such as Edge computing are readily available today, the community is moving closer to futuristic goals, such as drones over Internet Protocol [9]. Though such technologies eradicate the need for carrying heavy processors onboard, they introduce new challenges to control.

One of the fundamental challenges with the quadrotor control in these applications is the time delays arising from several sources, such as the communication medium, the sensors, the computational units, and the actuators. These individual delays accumulate to unpredictable time-varying closed-loop delays and thus affect the overall stability of the quadrotor dramatically. Another inherent challenge to quadrotor control is model uncertainties, due to its complex dynamics and unknown payloads. In these cases, parameter estimation becomes nearly impossible, especially when the quadrotor carries different unknown payloads. Along with the design, external factors, such as the wind, would destabilize the system. Hence, robustness towards delays and parametric uncertainties becomes inevitable in practical applications.

The aerial robotics community has comprehensively covered various aspects of uncertainties using robust control techniques for quadrotors. The controller [10] provides a pipeline for navigation and tracking for quadrotors with robustness to disturbances using an optimal control approach, while [8] establishes the robustness using a sliding-mode tracking controller. Similarly, Ref. [11] proposes a control scheme robust to input uncertainties. However, all these works rely on the complete knowledge of the dynamics without parametric uncertainties or delays. Further, let us explore the controllers that handle parametric uncertainties in the model. The robust controllers [12–15] mentioned above require a decent estimate of the system's dynamics. However, they introduce chattering in the dynamics when there is significant variation due to uncertainties. Several adaptive controllers are proposed to overcome the chattering issue. They are mostly classified into controllers [16–22] that require an a priori on the model of the system and the ones [23,24] that do not require any knowledge of the system. For the interested readers, a review of quadrotor UAVs in the aspects of their applications, architectural design, and control algorithms are presented in [25] for further reading. However, none of these controllers address the problem of delays in the closed loop. Thus, let us examine the effects of delays and the existing control approaches to tackle them.

Time delays have posed exciting challenges to the control research. Delays can cause different effects leading to instability in the closed-loop system depending on how they contribute to the system [26–30]. An overview of modelling a delayed system and the methods to control the system are surveyed in these works [31–33].

The control approaches for delayed systems mostly rely on the linear model of the dynamics to predict and compensate for the effects of delays. The controllers [34–36] designed to tackle the time-varying delays in the quadrotor control use a linear model of the dynamics and require the knowledge of the instantaneous delays. Linear approximation of the quadrotor's dynamics constraints it to comply with small angle assumptions, which reduces the dexterity of the quadrotor, causing instability in the presence of perturbations. Thus, nonlinear controllers tackling time delays are explored.

Since control of UAVs in the presence of time delays is not well-addressed to the best of the authors' knowledge, let us look into commonly addressed Euler–Lagrangian (EL) systems with delays. Ref. [37] proposes a robust observer-based sliding mode controller for bilateral latency, while an adaptive controller for the same application is proposed in [38]. A robust controller for a marine robot with input delay is proposed in [39]. Though these controllers provide a solution to input delays, they still require the knowledge of the upper bound of the derivatives of the delays, which is infeasible in real-time. These control approaches would not be suitable for the underactuated dynamics of a quadrotor with unknown and unbounded modeling uncertainties in the presence of unknown and varying time delays. Therefore, we proceed with the time delay approach similar to the one specified in [40], which requires neither a priori knowledge of the bounds nor the structure of parametric uncertainty. The controller also tackles unknown time-varying delays without any constraints on its derivative.

From these observations, we infer that an adaptive robust controller for an underactuated quadrotor UAV that tackles both unknown parametric uncertainties and time-varying delays is still missing in the literature. Towards this direction, the following contributions are proposed.

The first contribution of this work is a novel adaptive control technique for a quadrotor UAV for tracking any arbitrary 3D-position and yaw trajectories considering six DoF dynamics. The quadrotor is modeled using partly decoupled underactuated dynamics to consider perturbations and delays in both actuated and non-actuated dynamics. The controller needs neither the complete knowledge of the dynamics nor the upper bound of the uncertainties. Hence, it is robust to unknown modeling uncertainties and external disturbances. The adaptive law ensures that the switching gain does not increase monotonically. The second contribution is that the controller is robust to unknown time-varying delays with the knowledge only of the upper bound of the time delay without any constraint on

the derivatives of the delay. The controller's performance is verified using a simulated UAV with varying delays and parametric uncertainties. The stability analysis is performed using a Lyapunov-like method in the sense that the closed-loop trajectories of the system are uniformly ultimately bounded (UUB) (cf. [41] for the definition of UUB).

The rest of the article is organized as follows. Section 2 introduces the partly decoupled dynamics of the six-DoF quadrotor with uncertainties and delays, while Section 3 proposes the control structure of the quadrotor, Section 4 shows the implementation results of the proposed controller on a simulated quadrotor and Section 5 concludes the contributions made in the work and gives an insight of the future works. The stability analysis section is separately mentioned in Appendix A.

*Notations and Preliminaries*

The following notations are used throughout the article: any variable $\rho$ with a subscript $\mathbf{p}, \mathbf{q}$ as $\rho_{\mathbf{p}}, \rho_{\mathbf{q}}$ represents that the variable belongs to the position and the attitude sub-dynamics, respectively. Any variable $\rho$ delayed by an amount $h$ as $\rho(t - h)$ would be denoted as $\rho_h$; $\lambda_{min}(.)$ and $||.||$ represents the minimum eigenvalue and Euclidean norm of the argument, respectively, while $\mathbf{I}$ represents the identity matrix of the appropriate dimension. From the definition of Young's inequality, for any nonzero vectors $\mathbf{v}_1, \mathbf{v}_2$, there exists a constant $\beta > 0$ and a positive definite matrix $\mathbf{D} > 0$, such that,

$$-2\mathbf{v}_1^T\mathbf{v}_2 \leq \beta\mathbf{v}_1^T\mathbf{D}^{-1}\mathbf{v}_1 + \frac{1}{\beta}\mathbf{v}_2^T\mathbf{D}\mathbf{v}_2. \tag{1}$$

## 2. Quadrotor Dynamics

A quadrotor is a non-linear system that can be modelled using Euler–Lagrangian (EL) dynamics. The states of the quadrotor evolve based on the current states and the commanded inputs. The overall system delay propagates through the control loop and reflects in the control signal. Hence, the system can be modelled as a non-linear system with a delayed input, as given in (2).

$$m\ddot{\mathbf{p}}(t) + \mathbf{g} + \mathbf{d}_{\mathbf{p}}(t) = \boldsymbol{\tau}_{\mathbf{p}}(t - h(t)) \tag{2a}$$

$$\mathbf{J}(\mathbf{q}, t)\ddot{\mathbf{q}}(t) + \mathbf{C}(\mathbf{q}, \dot{\mathbf{q}}, t)\dot{\mathbf{q}}(t) + \mathbf{d}_{\mathbf{q}}(t) = \boldsymbol{\tau}_{\mathbf{q}}(t - h(t)) \tag{2b}$$

$$\boldsymbol{\omega}_{\mathbf{p}}(t) = \mathbf{R}_{\mathbf{B}}^{\mathbf{W}}(t)\mathbf{U}(t) \tag{2c}$$

where $m$ is the total mass of the system; $\mathbf{p}(t) \triangleq \begin{bmatrix} x(t) & y(t) & z(t) \end{bmatrix}^T \in \mathbb{R}^3$ is the position of the centre of mass of the quadrotor in the Earth-fixed frame at time, $t$; $\mathbf{q}(t) \triangleq \begin{bmatrix} \phi(t) & \theta(t) & \psi(t) \end{bmatrix}^T \in \mathbb{R}^3$ is the attitude vector consisting of the roll ($\phi$), pitch ($\theta$) and yaw ($\psi$) angles; $\mathbf{g} \triangleq \begin{bmatrix} 0 & 0 & mg \end{bmatrix}^T \in \mathbb{R}^3$, where $g$ is the acceleration due to gravity in the $z$-direction; $\mathbf{J}(\mathbf{q}, t) \in \mathbb{R}^{3 \times 3}$ is the inertia matrix; $\mathbf{C}(\mathbf{q}, \dot{\mathbf{q}}, t) \in \mathbb{R}^{3 \times 3}$ is the Coriolis matrix and the vectors $\mathbf{d}_{\mathbf{p}}, \mathbf{d}_{\mathbf{q}} \in \mathbb{R}^3$ represent the effect of the external disturbances (e.g., wind, gust), while $h(t)$ is the unknown time varying input-delay, $\boldsymbol{\tau}_{\mathbf{q}} \triangleq \begin{bmatrix} u_2(t) & u_3(t) & u_4(t) \end{bmatrix}^T \in \mathbb{R}^3$ denotes the control inputs for roll, pitch and yaw; $\boldsymbol{\tau}_{\mathbf{p}}(t) \in \mathbb{R}^3$ is the generalized control input for position tracking in Earth-fixed frame, with $\mathbf{U}(t) \triangleq \begin{bmatrix} 0 & 0 & u_1(t) \end{bmatrix}^T \in \mathbb{R}^3$ being the force vector in body-fixed frame and $\mathbf{R}_{B}^{W} \in \mathbb{R}^{3 \times 3}$ being the $Z - Y - X$ Euler angle rotation matrix describing the rotation from the body-fixed coordinate frame to the Earth-fixed frame, given by:

$$\mathbf{R}_B^W = \begin{bmatrix} c_\psi c_\theta & c_\psi s_\theta s_\phi - s_\psi c_\phi & c_\psi s_\theta c_\phi + s_\psi s_\phi \\ s_\psi c_\theta & s_\psi s_\theta s_\phi + c_\psi c_\phi & s_\psi s_\theta c_\phi - c_\psi s_\phi \\ -s_\theta & s_\phi c_\theta & c_\theta c_\phi \end{bmatrix}, \tag{3}$$

where $c_{(\cdot)}, s_{(\cdot)}$ and denote $\cos{(\cdot)}, \sin{(\cdot)}$ respectively. The control inputs are mapped to the rotor velocities using (4).

$$\begin{bmatrix} u_1 \\ u_2 \\ u_3 \\ u_4 \end{bmatrix} = \begin{bmatrix} C_T & C_T & C_T & C_T \\ 0 & lC_T & 0 & -lC_T \\ -lC_T & 0 & lC_T & 0 \\ -C_T C_M & C_T C_M & -C_T C_M & C_T C_M \end{bmatrix} \begin{bmatrix} \omega_1^2 \\ \omega_2^2 \\ \omega_3^2 \\ \omega_4^2 \end{bmatrix} \tag{4}$$

where $C_T$ and $C_M$ are the thrust constant and the moment constant, respectively, and $l$ is the arm-length as indicated in Figure 1).

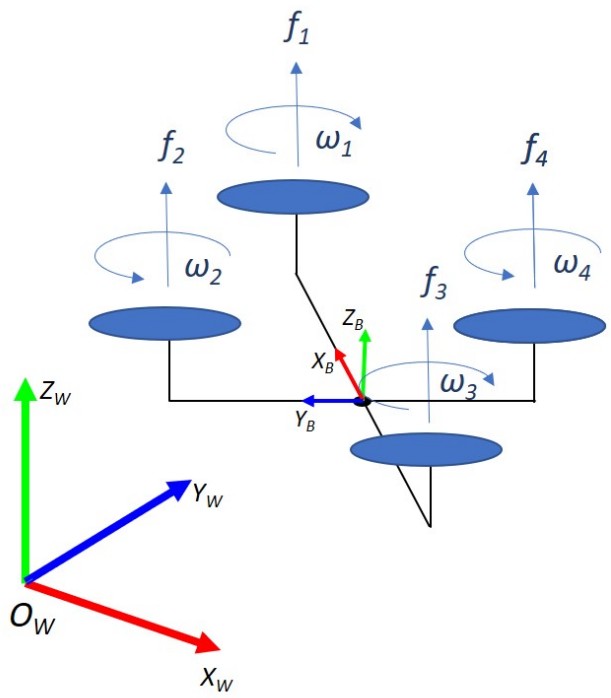

**Figure 1.** A schematic of the considered quadrotor with the related coordinate frames.

**Remark 1.** *To completely exploit the quadrotor's potential, it has to follow a trajectory in the three-dimensional space (x, y, z) along with the yaw angle (ψ). Though the roll and pitch angles are controlled, they are only utilized to achieve the desired motion in the 3-D space rather than following their own trajectory. Hence, we follow the partly decoupled tracking design (cf. [42]), rather than the completely decoupled design [19,43–45].*

We assume the following conditions about the dynamics and the desired outputs for the ease of control design.

**Assumption 1.** *By splitting the dynamic parameters into $m = \hat{m} + \Delta m$, $\mathbf{J} = \hat{\mathbf{J}} + \Delta \mathbf{J}$, $\mathbf{C} = \hat{\mathbf{C}} + \Delta \mathbf{C}$, we can approximate the dynamics into a sum of nominal values (.) and unknown uncertainties (Δ.). Let us assume that the uncertainties and disturbances are upper bounded by $|\Delta m| \leq \overline{m}$, $\underline{j} \leq ||\Delta \mathbf{J}|| \leq \overline{j}$, $||\Delta \mathbf{C}|| \leq \overline{c}$, $||\mathbf{d_p}|| \leq \overline{d_p}$ and $||\mathbf{d_q}|| \leq \overline{d_q}$, where $\overline{m}$, $\overline{j}$, $\overline{c}$, $\overline{d_p}$ and $\overline{d_q}$ are unknown scalars.*

**Remark 2.** *In practical applications, the nominal values of an unloaded quadrotor can be approximated using standard methods. Though the uncertainties of payload and the overall system are unknown, the quadrotors' maximum allowable limits are decided while choosing the hardware.*

**Assumption 2.** *The desired trajectory in position and yaw, given by $\left[x^d(t), y^d(t), z^d(t), \psi^d(t)\right]$ is smooth and feasible.*

The term "feasible" in the Assumption 2 refers to the feasibility of achieving the corresponding actuation for the calculated control inputs, which are dependant on the state errors. This assumption necessitates that the desired trajectories are bounded, which is a standard assumption in the UAV controller design [46].

**Control Problem:** Design a controller for a quadrotor to track a desired trajectory (cf. Assumption 2) with uncertainties in the dynamics and disturbances (cf. Assumption 1) in the presence of unknown time-varying delays.

## 3. Controller Design

The controller is designed to have an outer loop for position dynamics and an inner loop for attitude dynamics (cf. Figure 2). The outer loop takes the desired position as an input and yields the desired linear forces. The linear forces and the desired yaw angle produce the desired orientation for the inner loop.

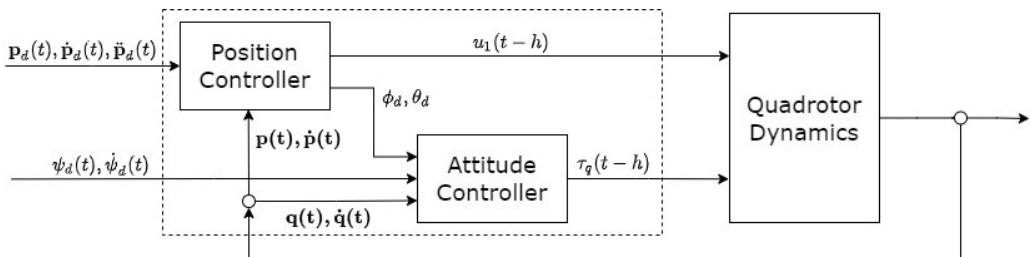

**Figure 2.** A schematic of the quadrotor control system. The control input is delayed by a time-varying function $h(t)$.

### 3.1. Position Control

Let $\mathbf{e_{1p}}(t) \triangleq \mathbf{p}^d(t) - \mathbf{p}(t)$ be the tracking error. The control input is designed as:

$$\boldsymbol{\tau_p} = \hat{m}\mathbf{u_p} + \hat{m}\mathbf{g}, \tag{5}$$

$$\mathbf{u_p} = \hat{\mathbf{u}}_\mathbf{p} + \Delta\mathbf{u_p} \tag{6}$$

The nominal control is designed as

$$\hat{\mathbf{u}}_\mathbf{p} = \ddot{\mathbf{p}}^d + \mathbf{K_{1p}}\mathbf{e_{1p}} + \mathbf{K_{2p}}\dot{\mathbf{e}}_{1p}, \tag{7}$$

where $\mathbf{K_{1p}}, \mathbf{K_{2p}} \in \mathbb{R}^3$, are two user-defined positive definite gain matrices. The error dynamics can be derived by substituting (7) and (6) in (2a),

$$\ddot{\mathbf{e}}_{1p} = -\mathbf{K_{1p}}\mathbf{e_{1ph}} - \mathbf{K_{2p}}\dot{\mathbf{e}}_{1ph} + \sigma_\mathbf{p} - \Delta\mathbf{u_{ph}}, \tag{8}$$

$$\sigma_\mathbf{p} = (1 - \frac{\hat{m}}{m})\mathbf{u_{ph}} + \frac{(m - \hat{m})}{m}\mathbf{g} + \ddot{\mathbf{p}}^d - \ddot{\mathbf{p}}_h^d + \mathbf{d_p} \tag{9}$$

where $\sigma_\mathbf{p}$ denotes overall uncertainty in the position control. Furthermore, by defining a vector $\mathbf{e_p} = \begin{bmatrix}\mathbf{e_{1p}} & \dot{\mathbf{e}}_{1p}\end{bmatrix}^T$, (8) can be represented as

$$\dot{\mathbf{e}}_\mathbf{p} = \mathbf{A_{1p}}\mathbf{e_p} + \mathbf{B_{1p}}\mathbf{e_{1p}} + \mathbf{B_p}(-\Delta\mathbf{u_{ph}} + \sigma_\mathbf{p}) \tag{10}$$

where $\mathbf{A}_{1p} \triangleq \begin{bmatrix} \mathbf{0} & \mathbf{I} \\ \mathbf{0} & \mathbf{0} \end{bmatrix}$, $\mathbf{B}_{1p} \triangleq \begin{bmatrix} \mathbf{0} & \mathbf{0} \\ -\mathbf{K_{1p}} & -\mathbf{K_{2p}} \end{bmatrix}$, $\mathbf{B_p} \triangleq \begin{bmatrix} \mathbf{0} & \mathbf{I} \end{bmatrix}^T$. Further, using the relation

$$\mathbf{e_{ph}} = \mathbf{e}(t)_\mathbf{p} - \int_{-h}^{0} \mathbf{e_p}(t + \delta)d\delta \tag{11}$$

the (10) can be rewritten as

$$\dot{\mathbf{e}}_{\mathbf{p}} = \mathbf{A}_{\mathbf{p}}\mathbf{e}_{\mathbf{p}} - \mathbf{B}_{1\mathbf{p}} \int_{-h}^{0} \mathbf{e}_{\mathbf{p}}(t+\delta)d\delta \quad + \mathbf{B}_{\mathbf{p}}(-\Delta\mathbf{u}_{\mathbf{p}h} + \sigma_{\mathbf{p}}), \tag{12}$$

where $\mathbf{A}_{\mathbf{p}} = \mathbf{A}_{1\mathbf{p}} + \mathbf{B}_{1\mathbf{p}}$ is in Hurwitz form. The filtered tracking error, $\mathbf{s}_{\mathbf{p}}$ is defined as,

$$\mathbf{s}_{p} \triangleq \mathbf{B}_{\mathbf{p}}{}^{T}\mathbf{P}_{\mathbf{p}}\mathbf{e}_{\mathbf{p}}, \tag{13}$$

where $\mathbf{P}_{\mathbf{p}} > \mathbf{0}$ is the solution to the Lyapunov equation $\mathbf{A}_{\mathbf{p}}{}^{T}\mathbf{P}_{\mathbf{p}} + \mathbf{P}_{\mathbf{p}}\mathbf{A}_{\mathbf{p}} = -\mathbf{Q}_{\mathbf{p}}$ for some $\mathbf{Q}_{\mathbf{p}} > \mathbf{0}$. The switching control input is designed as:

$$\Delta\mathbf{u}_{\mathbf{p}} = \begin{cases} \alpha_{\mathbf{p}}\hat{k}_{\mathbf{p}}\frac{\mathbf{s}_{\mathbf{p}}}{||\mathbf{s}_{\mathbf{p}}||} & \text{if } ||\mathbf{s}_{\mathbf{p}}|| \geq \varpi_{\mathbf{p}} \\ \alpha_{\mathbf{p}}\hat{k}_{\mathbf{p}}\frac{\mathbf{s}_{\mathbf{p}}}{\varpi_{\mathbf{p}}} & \text{if } ||\mathbf{s}_{\mathbf{p}}|| < \varpi_{\mathbf{p}} \end{cases}, \tag{14}$$

where $\alpha_{\mathbf{p}}$ is a user-defined positive gain, $\varpi_{\mathbf{p}} > 0$ is a scalar used as a saturation variable to avoid chattering along with $\hat{k}_{\mathbf{p}}$, which is an adaptive switching gain to tackle uncertainties, while the adaptive switching law for $\hat{k}_{\mathbf{p}}$ is given by:

$$\dot{\hat{k}}_{\mathbf{p}} = \begin{cases} ||\mathbf{s}_{\mathbf{p}}|| & \hat{k}_{\mathbf{p}} > \gamma_{\mathbf{p}}, \mathbf{s}_{\mathbf{p}}{}^{T}\dot{\mathbf{s}}_{\mathbf{p}} \geq 0 \\ -||\mathbf{s}_{\mathbf{p}}|| & \hat{k}_{\mathbf{p}} > \gamma_{\mathbf{p}}, \mathbf{s}_{\mathbf{p}}{}^{T}\dot{\mathbf{s}}_{\mathbf{p}} < 0 \\ \gamma_{p} & \hat{k}_{\mathbf{p}} \leq \gamma_{\mathbf{p}}, \end{cases} \tag{15}$$

where $\gamma_{\mathbf{p}} > 0$ is a small scalar to ensure $\hat{k}_{\mathbf{p}}$ is always positive. The adaptive law (15) increases or decreases the gain $\hat{k}_{\mathbf{p}}$ when the filtered error trajectories are moving away or are closer to 0, respectively.

### 3.2. Attitude Control

To achieve the attitude tracking control objective, the error in orientation/attitude is defined as [42]

$$\mathbf{e}_{\mathbf{q}} = ((\mathbf{R}^{d})^{T}\mathbf{R}_{B}^{W} - (\mathbf{R}_{B}^{W})^{T}\mathbf{R}^{d})^{v} \tag{16}$$

$$\dot{\mathbf{e}}_{\mathbf{q}} = \dot{\mathbf{q}} - (\mathbf{R}^{d})^{T}\mathbf{R}_{B}^{W}\dot{\mathbf{q}}^{d} \tag{17}$$

where $(.)^{v}$ represents *vee* map, which converts elements of $SO(3)$ to $\in \mathbb{R}^{3}$ [42] and $\mathbf{R}^{d}$ is the rotation matrix as in (3) evaluated at $(\phi^{d}, \theta^{d}, \psi^{d})$. Since the roll and pitch are used to navigate the quadrotor in XY-plane, the columns of the desired rotation matrix are derived from the desired force vector, as follows:

$$\mathbf{z}_{B} = \frac{\tau_{\mathbf{p}}}{||\tau_{\mathbf{p}}||} \tag{18a}$$

$$\mathbf{y}_{A} = \begin{bmatrix} -s_{\psi^{d}} & c_{\psi^{d}} & 0 \end{bmatrix}^{T} \tag{18b}$$

$$\mathbf{x}_{B} = \frac{\mathbf{y}_{A} \times \mathbf{z}_{B}}{||\mathbf{y}_{A} \times \mathbf{z}_{B}||} \tag{18c}$$

$$\mathbf{y}_{B} = \mathbf{z}_{B} \times \mathbf{x}_{B} \tag{18d}$$

where $\mathbf{y}_{A}$ is the $y$-axis of the intermediate coordinate frame $A$, $\mathbf{x}_{B}$, $\mathbf{y}_{B}$ and $\mathbf{z}_{B}$ form the desired body fixed coordinate frame (the columns of the desired rotation matrix).

**Assumption 3.** *The denominator of* (18a), $||\tau_{\mathbf{p}}|| \neq 0$.

Assumption 3 is a standard assumption while using the geometric control technique on the partly decoupled quadrotor dynamics (cf. [46]). However, in practical case, during the

rare event, when $||\tau_{\mathbf{p}}|| = 0$, the term $\mathbf{z}_B$ is chosen to be the previously calculated value. The inner loop controller is designed as follows:

$$\tau_{\mathbf{q}} = \hat{\mathbf{J}}\mathbf{u_q} + \hat{\mathbf{C}}\dot{\mathbf{q}}, \tag{19}$$

$$\mathbf{u_q} = \hat{\mathbf{u}}_{\mathbf{q}} + \Delta\mathbf{u_q} \tag{20}$$

The nominal control is designed as

$$\hat{\mathbf{u}}_{\mathbf{q}} = \ddot{\mathbf{q}}^d + \mathbf{K_{1q}}\mathbf{e_{1q}} + \mathbf{K_{2q}}\dot{\mathbf{e}}_{1q}, \tag{21}$$

where $\mathbf{K_{1q}}, \mathbf{K_{2q}} \in \mathbb{R}^3$, are two user-defined positive definite gain matrices. The error dynamics can be derived by substituting (21) and (20) in (2b),

$$\ddot{\mathbf{e}}_{1q} = -\mathbf{K_{1q}}\mathbf{e}_{1qh} - \mathbf{K_{2q}}\dot{\mathbf{e}}_{1qh} + \sigma_{\mathbf{q}} - \Delta\mathbf{u}_{\mathbf{q}h}, \tag{22}$$

where $\sigma_{\mathbf{q}} = (\mathbf{I} - \mathbf{J}^{-1}(\mathbf{q})\hat{\mathbf{J}}(\mathbf{q}))\mathbf{u}_{\mathbf{q}h} + \mathbf{J}^{-1}(\mathbf{q})(\mathbf{C}(\mathbf{q},\dot{\mathbf{q}})(\dot{\mathbf{q}}) - \hat{\mathbf{C}}(\mathbf{q}_h,\dot{\mathbf{q}}_h)(\dot{\mathbf{q}})) + \ddot{\mathbf{q}}^d - \ddot{\mathbf{q}}_h^d + \mathbf{d_q}$ denotes overall uncertainties in the position control. Further, (22) can be represented in state space using the vector $\mathbf{e_q} = \begin{bmatrix} \mathbf{e_{1q}} & \dot{\mathbf{e}}_{1q} \end{bmatrix}^T$ as,

$$\dot{\mathbf{e}}_{\mathbf{q}} = \mathbf{A_{1q}}\mathbf{e_q} + \mathbf{B_{1q}}\mathbf{e_{1q}} + \mathbf{B_q}(-\Delta\mathbf{u}_{\mathbf{q}h} + \sigma_{\mathbf{q}}), \tag{23}$$

where $\mathbf{A_{1q}} \triangleq \begin{bmatrix} \mathbf{0} & \mathbf{I} \\ \mathbf{0} & \mathbf{0} \end{bmatrix}$, $\mathbf{B_{1q}} \triangleq \begin{bmatrix} \mathbf{0} & \mathbf{0} \\ -\mathbf{K_{1q}} & -\mathbf{K_{2q}} \end{bmatrix}$, $\mathbf{B_q} \triangleq \begin{bmatrix} \mathbf{0} & \mathbf{I} \end{bmatrix}^T$. Further, using the relation $\mathbf{e_{qh}} = \mathbf{e}(t)_{\mathbf{q}} - \int_{-h}^{0} \mathbf{e_q}(t + \delta)d\delta$, (23) can be rewritten as,

$$\dot{\mathbf{e}}_{\mathbf{q}} = \mathbf{A_q}\mathbf{e_q} - \mathbf{B_{1q}} \int_{-h}^{0} \mathbf{e_q}(t + \delta)d\delta$$
$$+ \mathbf{B_q}(-\Delta\mathbf{u}_{\mathbf{q}h} + \sigma_{\mathbf{q}}), \tag{24}$$

where $\mathbf{A_q} = \mathbf{A_{1q}} + \mathbf{B_{1q}}$ is in Hurwitz form. The filtered tracking error, $\mathbf{s_q}$ is defined as:

$$\mathbf{s}_q \triangleq \mathbf{B_q}^T\mathbf{P_q}\mathbf{e_q}, \tag{25}$$

where $\mathbf{P_q} > 0$ is the solution to the Lyapunov equation $\mathbf{A_q}^T\mathbf{P_q} + \mathbf{P_q}\mathbf{A_q} = -\mathbf{Q_q}$ for some $\mathbf{Q_q} > \mathbf{0}$. The switching control input is designed as:

$$\Delta\mathbf{u_q} = \begin{cases} \alpha_{\mathbf{q}}\hat{k}_{\mathbf{q}}\dfrac{\mathbf{s_q}}{||\mathbf{s_q}||} & \text{if } ||\mathbf{s_q}|| \geq \varpi_{\mathbf{q}} \\ \alpha_{\mathbf{q}}\hat{k}_{\mathbf{q}}\dfrac{\mathbf{s_q}}{\varpi_{\mathbf{q}}} & \text{if } ||\mathbf{s_q}|| < \varpi_{\mathbf{q}}, \end{cases} \tag{26}$$

where $\alpha_{\mathbf{q}}$ is a user-defined positive gain, $\varpi_{\mathbf{q}} > 0$ is a scalar used as a saturation variable to avoid chattering along with $\hat{k}_{\mathbf{q}}$, which is an adaptive switching gain to tackle uncertainties. The adaptive switching law for $\hat{k}_{\mathbf{q}}$ is given by,

$$\dot{\hat{k}}_{\mathbf{q}} = \begin{cases} ||\mathbf{s_q}|| & \hat{k}_{\mathbf{q}} > \gamma_{\mathbf{q}}, \mathbf{s_q}^T\dot{\mathbf{s}}_{\mathbf{q}} \geq 0 \\ -||\mathbf{s_q}|| & \hat{k}_{\mathbf{q}} > \gamma_{\mathbf{q}}, \mathbf{s_q}^T\dot{\mathbf{s}}_{\mathbf{q}} < 0 \\ \gamma_q & \hat{k}_{\mathbf{q}} \leq \gamma_{\mathbf{q}}, \end{cases} \tag{27}$$

where $\gamma_{\mathbf{q}} > 0$ is a small scalar to ensure $\hat{k}_{\mathbf{q}}$ is always positive. The adaptive law (27) increases or decreases the gain $\hat{k}_{\mathbf{q}}$ when the filtered error trajectories are moving away or are closer to 0, respectively.

The proof of stability while using the proposed adaptive robust controller on an input-delayed quadrotor UAV is presented in Appendix A.

## 4. Simulation Results

To verify the performance of the controller, under uncertainties and time-delays, a test scenario is created using a MATLAB simulation where a quadrotor carrying a payload under external disturbances is modelled mathematically, while the proposed controller is tested with static and varying delays. The dynamics of the quadrotor is calculated based on the model proposed in [47] with an arm length of 24 cm. The mass of the quadrotor chassis and propeller is considered to be the nominal mass, $\hat{m} = 1$ kg. An additional 200 g along with the nominal mass is added considering the electronics system, which is the total mass of the unloaded quadrotor. Thus, the maximum mass of the quadrotor with the heavier payload is 1.9 kg and the inertia matrix for the same is calculated to be:

$$\mathbf{J} = \begin{bmatrix} 0.02352 & 0 & 0 \\ 0 & 0.02352 & 0 \\ 0 & 0 & 0.02704 \end{bmatrix},$$

Along with that, a time-varying disturbance is added to the dynamics as: $\mathbf{d_p} = cos(0.5t)\begin{bmatrix} 1 & 1 & 1 \end{bmatrix}^T$ and $\mathbf{d_q} = 0.01cos(0.01t)\begin{bmatrix} 1 & 1 & 1 \end{bmatrix}^T$. The control parameters are chosen to be $\mathbf{K_{1p}}, \mathbf{K_{2p}}, \mathbf{K_{1q}}, \mathbf{K_{2q}} = \mathbf{I}, \mathbf{Q_p}, \mathbf{Q_q} = \mathbf{I}$. The following values are used for nominal inertia and Coriolis matrices, $\hat{\mathbf{J}} = 0.01\mathbf{I}, \hat{\mathbf{C}} = \mathbf{0}$. The controller is experimented on three scenarios: (a) robustness to modelling uncertainties, (b) robustness to closed-loop delays and (c) robustness to closed-loop delays in the presence of uncertainties. The quadrotor is given the following trajectory to track in the scenarios:

$$x^d(t) = 1 + cos(0.5t) - exp(-t)$$
$$y^d(t) = sin(0.5t) - exp(-t)$$
$$z^d(t) = 2(1 - exp(-t)) + sin(0.1t)$$
$$\psi_d(t) = 0$$

### 4.1. Robustness to Modelling Uncertainties

The purpose of this scenario is to demonstrate that the controller is robust to unknown external disturbances and dynamic uncertainties. Thus, initially the controller is tested with the quadrotor with exact dynamic parameters and no external disturbances. Then, the same controller is tested with a model of a quadrotor carrying a small payload (0.3 kg) and external disturbances. Following that, the controller is tested with a model of a quadrotor carrying a heavier payload (0.7 kg) with the same disturbances. During this scenario, the closed-loop delay is set to be zero. The comparison of root mean squared error (RSME) values for position tracking for the scenario is tabulated in Table 1. Figure 3 shows the 3D plot of the quadrotors with different payloads along with the desired trajectory (blue). It can be observed that the effects of dynamic uncertainties and disturbances are minimized and the quadrotor follows the reference trajectory with only a minimal deviation.

**Table 1.** Position tracking performance comparison robustness to dynamic uncertainties.

| Payload Mass (kg) | RMS Position Error (m) | | |
|:---:|:---:|:---:|:---:|
| | $x$ | $y$ | $z$ |
| 0 | 0.0267 | 0.0388 | 0.0325 |
| 0.3 | 0.0800 | 0.0801 | 0.0606 |
| 0.7 | 0.1278 | 0.1242 | 0.1423 |

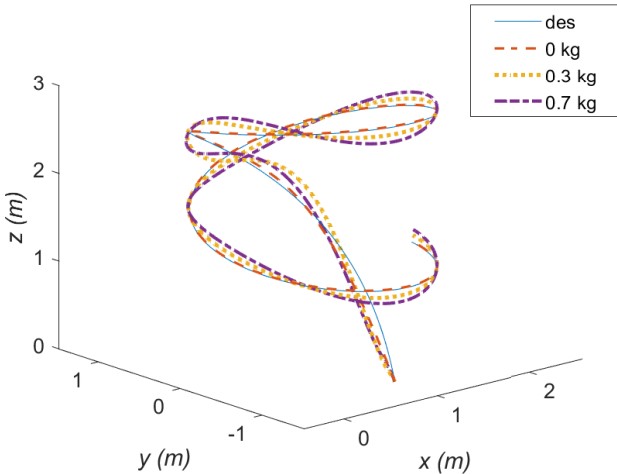

**Figure 3.** Robustness to uncertainty in dynamics: The plot shows the actual trajectories taken by the quadrotor with various payloads along with the desired trajectory without considering the delays.

### 4.2. Robustness to Closed-Loop Time-Delays

In this scenario, the controller is analyzed with a closed-loop delay in the system. The performance is compared with the no-delay system. The scenario is experimented on the quadrotor model without any dynamic uncertainties and disturbances. The controller is tested with static closed-loop delays of 0.2 s and 0.4 s, and a time-varying delay that varies between 0 s to 0.8 s. The time-varying delay function is as shown in the Figure 4. Figure 5 shows that all trajectories converge smoothly over the desired trajectory. The closeness between the RSME values for various delays in Table 2 prove the robustness of the controller to unknown time-delays.

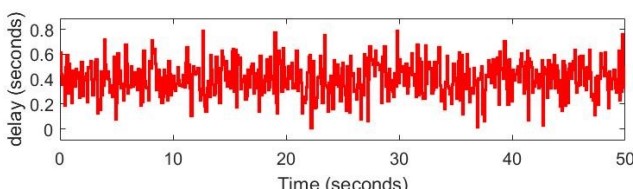

**Figure 4.** The input from the controller is delayed using a delay function with a time-varying delay. The delay varies from 0 s to 0.8 s with a mean of 0.4 s.

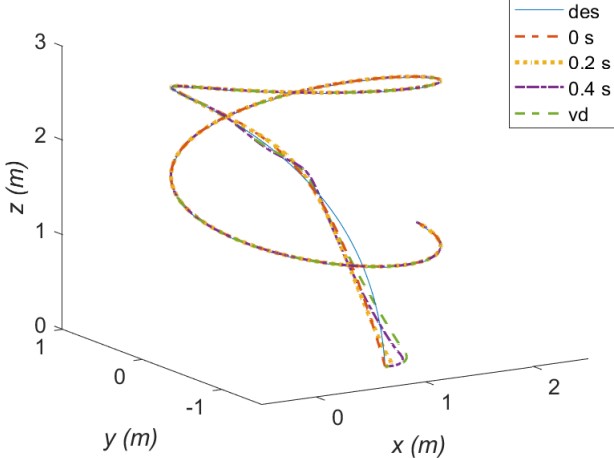

**Figure 5.** Robustness to delays: The controller is tested on a controller without any payload by only varying the delays in the system. The plot shows the actual trajectory of the quadrotor with constant and varying delays along with the desired trajectory.

**Table 2.** Position tracking performance comparison for robustness to time-delays.

| Delay (s) | RMS Position Error (m) | | |
|:---:|:---:|:---:|:---:|
| | $x$ | $y$ | $z$ |
| 0 | 0.0267 | 0.0388 | 0.0325 |
| 0.2 | 0.0570 | 0.0829 | 0.0844 |
| 0.4 | 0.0732 | 0.1060 | 0.1115 |
| Variable | 0.0701 | 0.1122 | 0.1228 |

*4.3. Robustness to Closed-Loop Time-Delays with Modelling Uncertainties*

In many applications, the quadrotor control is affected by both delays and uncertainties. Thus, in this scenario, the control performance is tested on the quadrotor model with different closed-loop delays in the presence of heavy disturbances and parametric uncertainties, as in the third case of first scenario. From the obtained results, it is observed from the RSME values, mentioned in Table 3, that the controller is simultaneously robust to both uncertainties and delays. Though the performance slightly deteriorates with high payload and large delays, the quadrotor is found to be stable and closely following the desired trajectory (cf. Figure 6).

**Table 3.** Position tracking performance comparison for robustness to time-delays with uncertainties.

| Delay (s) | RMS Position Error (m) | | |
|:---:|:---:|:---:|:---:|
| | $x$ | $y$ | $z$ |
| 0 | 0.1278 | 0.1242 | 0.1423 |
| 0.2 | 0.1547 | 0.1478 | 0.1857 |
| 0.4 | 0.2007 | 0.1905 | 0.2114 |
| Variable | 0.1999 | 0.1918 | 0.2091 |

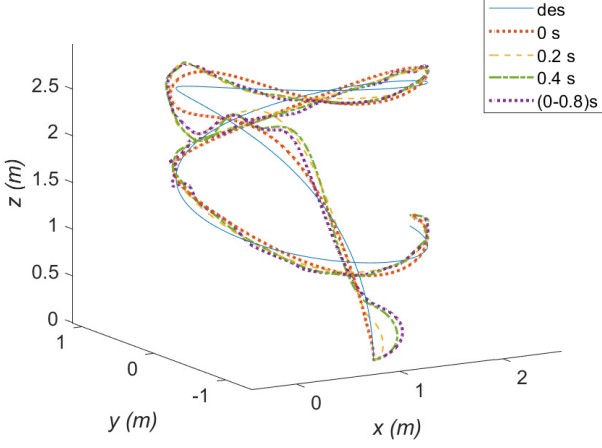

**Figure 6.** Robustness to delays with uncertainties: The plot shows the actual trajectories taken by the quadrotor with maximum payload in the presence of various constant and varying delays along with the desired trajectory.

**5. Conclusions and Future Works**

In this article, a novel adaptive robust controller is proposed for a quadrotor to track any arbitrary trajectory using the 6-DoF decoupled dynamics approach. The proposed control scheme considers perturbations in all the actuated and non-actuated sub-dynamics. Hence, it is robust to unknown uncertainties and external disturbances. It also tackles

the effects of time-varying closed-loop delays with only requiring the knowledge of the upper bounds of the delays. The controller is tested on a MATLAB simulation for three different scenarios to verify the performance. The stability of the closed-loop system using the proposed controller is analysed in the sense of UUB using a Lyapunov-like method.

As a future work, we would like to extend the theory by adding a tunable parameter in the adaptive laws, which regulate the rate of convergence. Another challenge that we are currently working on is to tackle the uncertainties with a priori knowledge of neither the system dynamics nor the bounds of the uncertainties.

**Author Contributions:** Conceptualization, V.N.S., S.S. and G.N.; methodology, V.N.S.; simulation and results, V.N.S.; validation, S.S. and G.N.; preparation of initial draft, V.N.S., S.S. and G.N.; modifications and revisions, V.N.S., S.S. and G.N. All authors have read and agreed to the published version of the manuscript.

**Funding:** This work has been partially funded by the European Unions Horizon 2020 Research and Innovation Programme AERO-TRAIN under the Grant Agreement No. 953454.

**Institutional Review Board Statement:** Not applicable.

**Informed Consent Statement:** Not applicable.

**Data Availability Statement:** Not applicable.

**Conflicts of Interest:** The authors declare that no conflicts of interest or competing interests are associated with this work.

## Appendix A. Stability Analysis

**Lemma A1.** *Under Assumptions 1–3, the overall uncertainties in the position and attitude dynamics are bounded by unknown scalars $k_\mathbf{p}, k_\mathbf{q}$, such that $||\sigma_\mathbf{p}|| \leq k_\mathbf{p}, ||\sigma_\mathbf{q}|| \leq k_\mathbf{q}$.*

**Proof.** Substituting the nominal and uncertain terms defined in Assumption 1, the values of $||\sigma_\mathbf{p}||, ||\sigma_\mathbf{q}||$ can be written as,

$$
\begin{aligned}
||\sigma_\mathbf{p}|| &\leq ||(\frac{\Delta m}{m})\mathbf{u}_{\mathbf{p}h} + (\frac{\Delta m}{m})\mathbf{g} + \ddot{\mathbf{p}}^d - \ddot{\mathbf{p}}_h^d + \mathbf{d}_\mathbf{p}|| \\
&\leq (\frac{\overline{m}}{\hat{m}})||\mathbf{u}_{\mathbf{p}h}|| + (\frac{\overline{m}}{\hat{m}})||\mathbf{g}|| + ||\ddot{\mathbf{p}}^d - \ddot{\mathbf{p}}_h^d|| + \overline{d}_\mathbf{p}
\end{aligned}
\tag{A1}
$$

$$
\begin{aligned}
||\sigma_\mathbf{q}|| &\leq ||\mathbf{J}^{-1}\Delta\mathbf{J}\mathbf{u}_{\mathbf{q}h} + \mathbf{J}^{-1}\Delta\mathbf{C}\dot{\mathbf{q}} + \ddot{\mathbf{q}}^d - \ddot{\mathbf{q}}_h^d + \mathbf{d}_\mathbf{q}|| \\
&\leq \overline{j}(\hat{\mathbf{J}})^{-1}||\mathbf{u}_{\mathbf{q}h}|| + \overline{c}(\hat{\mathbf{J}})^{-1}||\dot{\mathbf{q}}|| + ||\ddot{\mathbf{q}}^d - \ddot{\mathbf{q}}_h^d|| + \overline{d}_\mathbf{q}
\end{aligned}
\tag{A2}
$$

Since the desired trajectories are well-defined and bounded in accordance with Assumptions 2 and 3, the values in (A1), (A2) can be simplified as using the unknown scalars $k_\mathbf{p}, k_\mathbf{q}$,

$$
||\sigma_\mathbf{p}|| \leq k_\mathbf{p}
\tag{A3}
$$

$$
||\sigma_\mathbf{q}|| \leq k_\mathbf{q}
\tag{A4}
$$

□

To further proceed with the analysis, let us choose the Lyapunov function $V(\mathbf{e}_\mathbf{p}(t), \mathbf{e}_\mathbf{q}(t)) > 0$, $\forall t \geq 0$ given by,

$$
V(\mathbf{e}_\mathbf{p}, \mathbf{e}_\mathbf{q}) = V_p(\mathbf{e}_\mathbf{p}) + V_q(\mathbf{e}_\mathbf{q})
\tag{A5}
$$

$$
V_p(\mathbf{e}_\mathbf{p}) = \frac{1}{2}\mathbf{e}_\mathbf{p}^T\mathbf{P}_\mathbf{p}\mathbf{e}_\mathbf{p} + \frac{1}{2}(\hat{k}_\mathbf{p} - k_\mathbf{p})^2
\tag{A6}
$$

$$
V_q(\mathbf{e}_\mathbf{q}) = \frac{1}{2}\mathbf{e}_\mathbf{q}^T\mathbf{P}_\mathbf{q}\mathbf{e}_\mathbf{q} + \frac{1}{2}(\hat{k}_\mathbf{q} - k_\mathbf{q})^2
\tag{A7}
$$

The time derivatives of $V_p$, $V_q$ are simplified through Lemmas A2 and A3 respectively, to facilitate the stability analysis of the closed loop systems, as stated in Theorem A1.

**Lemma A2.** *Under the Assumptions 1–3, the time derivative of $V_p$ in (A6) can be simplified using (12) and (13) as,*

$$\dot{V}_p \leq -\frac{1}{2}\mathbf{e}_\mathbf{p}^T[\mathbf{Q}_\mathbf{p} - h\mathbf{E}_\mathbf{p}]\mathbf{e}_\mathbf{p} + \Gamma_p + \mathbf{s}_\mathbf{p}^T(-\Delta\mathbf{u}_\mathbf{p} + \sigma_p) + \mathbf{s}_\mathbf{p}^T\Theta_p + (\hat{k}_\mathbf{p} - k_\mathbf{p})\dot{\hat{k}}_\mathbf{p} \quad \text{(A8)}$$

*where*

$$\mathbf{E}_\mathbf{p} = \beta\mathbf{P}_\mathbf{p}\mathbf{B}_{1\mathbf{p}}[\mathbf{A}_{1\mathbf{p}}\mathbf{P}_\mathbf{p}^{-1}\mathbf{A}_{1\mathbf{p}}^T + \mathbf{B}_{1\mathbf{p}}\mathbf{P}_\mathbf{p}^{-1}\mathbf{B}_{1\mathbf{p}}^T + \mathbf{P}_\mathbf{p}^{-1}]\mathbf{B}_{1\mathbf{p}}^T\mathbf{P}_\mathbf{p} + \frac{2r}{\beta}\mathbf{P}_\mathbf{p}, \quad \text{(A9)}$$

$$\Gamma_p \geq \frac{1}{2\beta}||\int_{-h}^0[\phi_p(t+\delta) + \phi_p(t-h+\delta) + \sigma_{1\mathbf{p}}^T(t+\delta)\mathbf{B}_\mathbf{p}^T\mathbf{P}_\mathbf{p}\mathbf{B}_\mathbf{p}\sigma_{1\mathbf{p}}(t+\delta)]d\delta||, \quad \text{(A10)}$$

$$\Theta_p = \Delta\mathbf{u}_\mathbf{p} - \Delta\mathbf{u}_{\mathbf{p}h}, \quad \text{(A11)}$$

*assuming that within the delayed interval, the uncertainties are integrable (i.e., locally Lipschitz within the interval of delay), where $\varphi_p(\epsilon) \triangleq r(\hat{k}_\mathbf{p}(t) - k_\mathbf{p}(t))^2 - (\hat{k}_\mathbf{p}(\epsilon) - k_\mathbf{p}(\epsilon))^2$, $\sigma_{1\mathbf{p}} = -\Delta\mathbf{u}_{\mathbf{p}h} + \sigma_\mathbf{p}$.*

**Proof.** Using (12), (13), the time derivative of the Lyapunov candidate $V_p$ in (A6) is given by,

$$\dot{V}_p = \underbrace{-\frac{1}{2}\mathbf{e}_\mathbf{p}^T\mathbf{Q}_\mathbf{p}\mathbf{e}_\mathbf{p}}_{\eta_{p1}} \underbrace{-\mathbf{e}_\mathbf{p}^T\mathbf{P}_\mathbf{p}\mathbf{B}_{1\mathbf{p}}\int_{-h}^0\dot{\mathbf{e}}_p(t+\delta)d\delta + \mathbf{s}_\mathbf{p}^T(-\Delta\mathbf{u}_{\mathbf{p}h} + \sigma_\mathbf{p})}_{\eta_{p2}} + \underbrace{(\hat{k}_\mathbf{p} - k_\mathbf{p})\dot{\hat{k}}_\mathbf{p}}_{\eta_{p3}}, \quad \text{(A12)}$$

where $\eta_{p2}$ can be simplified using (10) as,

$$-\mathbf{e}_\mathbf{p}^T\mathbf{P}_\mathbf{p}\mathbf{B}_{1\mathbf{p}}\int_{-h}^0\dot{\mathbf{e}}_p(t+\delta)d\delta = -\int_{-h}^0\mathbf{e}_\mathbf{p}^T\mathbf{P}_\mathbf{p}\mathbf{B}_{1\mathbf{p}}[\mathbf{A}_{1p}\mathbf{e}_\mathbf{p}(t+\delta) + \mathbf{B}_{1\mathbf{p}}\mathbf{e}_\mathbf{p}(t-h+\delta)$$
$$+ \mathbf{B}_\mathbf{p}\sigma_{1\mathbf{p}}(t+\delta)]d\delta, \quad \text{(A13)}$$

The terms in (A13) can be further simplified using the inequality in (1) by choosing $\mathbf{D} = \mathbf{P}_\mathbf{p}$ as follows,

$$-\int_{-h}^0\mathbf{e}_\mathbf{p}^T\mathbf{P}_\mathbf{p}\mathbf{B}_{1\mathbf{p}}[\mathbf{A}_{1p}\mathbf{e}_\mathbf{p}(t+\delta)]d\delta \leq \frac{\beta}{2}\int_{-h}^0\mathbf{e}_\mathbf{p}^T\mathbf{P}_\mathbf{p}\mathbf{B}_{1\mathbf{p}}\mathbf{A}_{1p}\mathbf{P}_\mathbf{p}^{-1}\mathbf{A}_{1p}^T\mathbf{B}_{1\mathbf{p}}^T\mathbf{P}_\mathbf{p}^T\mathbf{e}_\mathbf{p}d\delta$$
$$+ \frac{1}{2\beta}\int_{-h}^0\mathbf{e}_\mathbf{p}^T(t+\delta)\mathbf{P}_\mathbf{p}\mathbf{e}_\mathbf{p}(t+\delta)d\delta \quad \text{(A14)}$$

$$-\int_{-h}^0\mathbf{e}_\mathbf{p}^T\mathbf{P}_\mathbf{p}\mathbf{B}_{1\mathbf{p}}\mathbf{B}_{1p}\mathbf{e}_\mathbf{p}(t-h+\delta)d\delta \leq \frac{\beta}{2}\int_{-h}^0\mathbf{e}_\mathbf{p}^T\mathbf{P}_\mathbf{p}\mathbf{B}_{1\mathbf{p}}\mathbf{B}_{1\mathbf{p}}\mathbf{P}_\mathbf{p}^{-1}\mathbf{B}_{1\mathbf{p}}^T\mathbf{B}_{1\mathbf{p}}^T\mathbf{P}_\mathbf{p}^T\mathbf{e}_\mathbf{p}d\delta$$
$$+ \frac{1}{2\beta}\int_{-h}^0\mathbf{e}_\mathbf{p}^T(t-h+\delta)\mathbf{P}_\mathbf{p}\mathbf{e}_\mathbf{p}(t-h+\delta)d\delta \quad \text{(A15)}$$

$$-\int_{-h}^0\mathbf{e}_\mathbf{p}^T\mathbf{P}_\mathbf{p}\mathbf{B}_{1\mathbf{p}}\mathbf{B}_\mathbf{p}\sigma_{1\mathbf{p}}(t+\delta)d\delta \leq \frac{\beta}{2}\int_{-h}^0\mathbf{e}_\mathbf{p}^T\mathbf{P}_\mathbf{p}\mathbf{B}_{1\mathbf{p}}\mathbf{P}_\mathbf{p}^{-1}\mathbf{B}_{1\mathbf{p}}^T\mathbf{P}_\mathbf{p}^T\mathbf{e}_\mathbf{p}d\delta$$
$$+ \frac{1}{2\beta}\int_{-h}^0\sigma_{1\mathbf{p}}^T(t+\delta)\mathbf{B}_\mathbf{p}^T\mathbf{P}_\mathbf{p}\mathbf{B}_\mathbf{p}\sigma_{1\mathbf{p}}(t+\delta)d\delta$$
$$\leq \frac{h}{2}\mathbf{e}_p^T[\beta\mathbf{P}_\mathbf{p}\mathbf{B}_{1\mathbf{p}}\mathbf{P}_\mathbf{p}^{-1}\mathbf{B}_{1\mathbf{p}}^T\mathbf{P}]\mathbf{e}_\mathbf{p}$$
$$+ \frac{1}{2\beta}\int_{-h}^0\sigma_{1\mathbf{p}}^T(t+\delta)\mathbf{B}_\mathbf{p}^T\mathbf{P}_\mathbf{p}\mathbf{B}_\mathbf{p}\sigma_{1\mathbf{p}}(t+\delta)d\delta, \quad \text{(A16)}$$

To further simplify (A14), (A15), let us recall the inequality from the Razumikhin theorem [48],

$$V_p(\mathbf{e_p}(\epsilon)) < rV_p(\mathbf{e_p}(t)), \qquad\qquad t - 2h \le \epsilon \le t \qquad\qquad \text{(A17)}$$

$$\implies \mathbf{e_p}^T(\epsilon)\mathbf{P_p}\mathbf{e_p}(\epsilon) < r\mathbf{e}_p^T(t)\mathbf{P_p}\mathbf{e}_p(t) + \varphi_p(\epsilon), \qquad\qquad \text{(A18)}$$

where $r > 1$ is some constant. Using (A18), the second terms in (A14) and (A15) can be simplified as,

$$-\int_{-h}^0 \mathbf{e_p}^T\mathbf{P_p}\mathbf{B_{1p}}[\mathbf{A}_{1p}\mathbf{e_p}(t+\delta)]d\delta \le \frac{h}{2}\mathbf{e_p}^T[\beta\mathbf{P_p}\mathbf{B_{1p}}\mathbf{A_{1p}}\mathbf{P}_p^{-1}\mathbf{A_{1p}}^T\mathbf{B_{1p}}^T\mathbf{P_p} + \frac{r}{2\beta}\mathbf{P_p}]\mathbf{e_p}$$

$$+ \frac{1}{\beta}\int_{-h}^0 \varphi_p(t+\delta)d\delta \qquad\qquad \text{(A19)}$$

$$-\int_{-h}^0 \mathbf{e_p}^T\mathbf{P_p}\mathbf{B_{1p}}\mathbf{B_{1p}}\mathbf{e_p}(t-h+\delta)d\delta \le \frac{h}{2}\mathbf{e_p}^T[\beta\mathbf{P_p}\mathbf{B_{1p}}\mathbf{B_{1p}}\mathbf{P_p}^{-1}\mathbf{B_{1p}}^T\mathbf{B_{1p}}^T\mathbf{P_p} + \frac{r}{2\beta}\mathbf{P_p}]\mathbf{e_p}$$

$$+ \frac{1}{2\beta}\int_{-h}^0 \varphi_p(t-h+\delta)d\delta \qquad\qquad \text{(A20)}$$

From the simplifications of the terms in $\eta_{p2}$ in (A19), (A20) and (A16), the derivative of $V_p$ in (A12) can be rewritten as,

$$\dot{V}_p \le -\frac{1}{2}\mathbf{e_p}^T[\mathbf{Q_p} - h\mathbf{E_p}]\mathbf{e_p} + \Gamma_p + \mathbf{s_p}^T(-\Delta\mathbf{u_p} + \sigma_p) + \mathbf{s_p}^T\Theta_p + (\hat{k}_\mathbf{p} - k_\mathbf{p})\dot{\hat{k}}_\mathbf{p} \qquad \text{(A21)}$$

□

**Lemma A3.** *Under the Assumptions 1–3, the time derivative of $V_q$ in (A7) can be simplified using (24) and (25) as,*

$$\dot{V}_q \le -\frac{1}{2}\mathbf{e_q}^T[\mathbf{Q_q} - h\mathbf{E_q}]\mathbf{e_q} + \Gamma_q + \mathbf{s_q}^T(-\Delta\mathbf{u_q} + \sigma_q) + \mathbf{s_q}^T\Theta_q + (\hat{k}_\mathbf{q} - k_\mathbf{q})\dot{\hat{k}}_\mathbf{q} \qquad \text{(A22)}$$

*where*

$$\mathbf{E_q} = \beta\mathbf{P_q}\mathbf{B_{1q}}[\mathbf{A_{1q}}\mathbf{P_q}^{-1}\mathbf{A_{1q}}^T + \mathbf{B_{1q}}\mathbf{P_q}^{-1}\mathbf{B_{1q}}^T + \mathbf{P_q}^{-1}]\mathbf{B_{1q}}^T\mathbf{P_q} + \frac{2r}{\beta}\mathbf{P_q}, \qquad \text{(A23)}$$

$$\Gamma_q \ge \frac{1}{2\beta}||\int_{-h}^0 [\phi_q(t+\delta) + \phi_q(t-h+\delta) + \sigma_{1\mathbf{q}}^T(t+\delta)\mathbf{B_q}^T\mathbf{P_q}\mathbf{B_q}\sigma_{1\mathbf{q}}(t+\delta)]d\delta||, \qquad \text{(A24)}$$

$$\Theta_q = \Delta\mathbf{u_q} - \Delta\mathbf{u_{qh}}, \qquad \text{(A25)}$$

*assuming that within the delayed interval, the uncertainties are integrable (i.e., locally Lipschitz within the interval of delay), where $\varphi_q(\epsilon) \triangleq r(\hat{k}_\mathbf{q}(t) - k_\mathbf{q}(t))^2 - (\hat{k}_\mathbf{q}(\epsilon) - k_\mathbf{q}(\epsilon))^2$, $\sigma_{1\mathbf{q}} = -\Delta\mathbf{u_{qh}} + \sigma_\mathbf{q}$.*

**Proof.** Since, $V_q$ in (A7), the uncertainty bound in (A4), the control law in (20) and the adaptive law in (27) for the attitude sub-dynamics (2) are similar to those of the position sub-dynamics in (A3), (A6), (6) and (15) respectively, by following a similar procedure from (A12) to (A41) (as in Lemma A2) and arrive at the relationship,

$$\dot{V}_q \le -\frac{1}{2}\mathbf{e_q}^T[\mathbf{Q_q} - h\mathbf{E_q}]\mathbf{e_q} + \Gamma_q + \mathbf{s_q}^T(-\Delta\mathbf{u_q} + \sigma_q) + \mathbf{s_q}^T\Theta_q + (\hat{k}_\mathbf{q} - k_\mathbf{q})\dot{\hat{k}}_\mathbf{q}. \qquad \text{(A26)}$$

□

**Assumption 4.** *The unknown time-varying delay, h is bounded, such that, $h(t) < \bar{h}$, $\forall t \ge 0$, where $\bar{h} > 0$ is a scalar.*

**Lemma A4.** *Under Assumption 4, the terms* $\Lambda_i \triangleq [\mathbf{Q}_i - h(t)\mathbf{E}_i]$, $i \in \{\mathbf{p}, \mathbf{q}\}$ *are positive definite* $\forall t \geq 0$.

**Proof.** From the upper bounds of $\dot{V}_p$, $\dot{V}_q$ in (A21) and (A26) respectively, in Lemmas A2 and A3, the following relationship should hold for the matrices, $\Lambda_i \triangleq [\mathbf{Q}_i - h(t)\mathbf{E}_i]$, $i \in \{\mathbf{p}, \mathbf{q}\}$ to be positive definite $\forall t \geq 0$,

$$h(t) < \mathbf{Q_p}\mathbf{E_p}^{-1}$$
$$< \frac{\lambda_{min}(\mathbf{Q_p})}{||\mathbf{E_p}||} \tag{A27}$$

$$h(t) < \mathbf{Q_q}\mathbf{E_q}^{-1}$$
$$< \frac{\lambda_{min}(\mathbf{Q_q})}{||\mathbf{E_q}||}. \tag{A28}$$

Further, by combining the inequality conditions in (A27) and (A28), and using $\overline{h}$ from the Assumption 4, we have,

$$\overline{h} < \frac{\min_{i \in \mathbf{p}, \mathbf{q}}\{\lambda_{min}(\mathbf{Q}_i)\}}{\max_{i \in \mathbf{p}, \mathbf{q}}\{||\mathbf{E}_i||\}}. \tag{A29}$$

By choosing appropriate positive definite matrices, $\mathbf{Q_p}, \mathbf{Q_q}$, and parameters $\mathbf{K_{1p}}, \mathbf{K_{2p}}$, $\mathbf{K_{1q}}, \mathbf{K_{2q}}$, $r$, and $\beta$, the inequality constraint in (A29) can be satisfied for any value of The inequality constraint in (A29) can be satisfied by choosing appropriate positive definite matrices, $\mathbf{Q_p}, \mathbf{Q_q}$, and parameters $\mathbf{K_{1p}}, \mathbf{K_{2p}}, \mathbf{K_{1q}}, \mathbf{K_{2q}}$, $r$, and $\beta$. Hence, the □

**Theorem A1.** *Under the Assumptions 1–4, the closed loop trajectories of the system defined in (2) using the control laws (6) and (20), and the adaptive laws (15) and (27) are uniformly ultimately bounded (cf. Def. 4.6 in [41] for the definition of UUB).*

**Proof.** From Assumption 4, the expression (A21) can be simplified using the positive definite matrix, $\Lambda_p \triangleq [\mathbf{Q_p} - h\mathbf{E_p}] > 0$ as,

$$\dot{V}_p \leq -\frac{1}{2}\mathbf{e_p}^T\Lambda_p\mathbf{e_p} + \Gamma_p + \mathbf{s_p}^T(-\Delta\mathbf{u_p} + \sigma_p) + \mathbf{s_p}^T\Theta_p + (\hat{k}_\mathbf{p} - k_\mathbf{p})\dot{\hat{k}}_\mathbf{p} \tag{A30}$$

$$\leq -\frac{1}{2}\mathbf{e_p}^T\lambda_{min}(\Lambda_p)\mathbf{e_p} + \Gamma_p + \mathbf{s_p}^T(-\Delta\mathbf{u_p} + k_\mathbf{p}) + (\hat{k}_\mathbf{p} - k_\mathbf{p})\dot{\hat{k}}_\mathbf{p} + ||\mathbf{s_p}||||\Theta_p||. \tag{A31}$$

From (A6), the upper bound of $V_p$ can be defined as,

$$V_p \leq \frac{1}{2}\lambda_{max}\mathbf{P_p}||\mathbf{e_p}||^2 + \frac{1}{2}(\hat{k}_\mathbf{p} - k_\mathbf{p})^2. \tag{A32}$$

Using the relationships in (A31) and (A32), it can be observed that,

$$\dot{V}_p \leq -\varrho_p V_p + \Gamma_p + \mathbf{s_p}^T(-\Delta\mathbf{u_p} + k_\mathbf{p}) + (\hat{k}_\mathbf{p} - k_\mathbf{p})\dot{\hat{k}}_\mathbf{p} + ||\mathbf{s_p}||||\Theta_p|| \tag{A33}$$

where $\varrho_p \triangleq \frac{\lambda_{min}(\Lambda_p)}{min\{\lambda_{max}(\mathbf{P_p})/2, 1/2\}}$ is a positive scalar. Further, let us analyze (A31) under the different possible cases:

**Case 1:** $\hat{k}_\mathbf{p} > \gamma_\mathbf{p}$, $\mathbf{s_p}^T\dot{\mathbf{s}}_\mathbf{p} > 0$, $||\mathbf{s_p}|| \geq \varpi_\mathbf{p}$

Substituting the values of $\Delta\mathbf{u_p}, \dot{\hat{k}}_\mathbf{p}$ from the control law (20) and adaptive law (27) in (A31) implies,

$$\dot{V}_p \leq -\frac{1}{2}\lambda_{min}(\Lambda_p)||\mathbf{e_p}||^2 + \Gamma_p + ||\mathbf{s_p}||||\Theta_p|| + \left(-\alpha_{\mathbf{p}}\hat{k}_{\mathbf{p}}\frac{\mathbf{s_p^T s_p}}{||\mathbf{s_p}||} + k_{\mathbf{p}}||\mathbf{s_p}||\right) + (\hat{k}_{\mathbf{p}} - k_{\mathbf{p}})||\mathbf{s_p}||$$

$$\leq -\frac{1}{2}\lambda_{min}(\Lambda_p)||\mathbf{e_p}||^2 + \Gamma_p + ||\mathbf{s_p}||||\Theta_p|| - \hat{k}_{\mathbf{p}}(\alpha_{\mathbf{p}} - 1)||\mathbf{s_p}||. \tag{A34}$$

Simply by choosing $\alpha_p > 1$, it can be observed that $\dot{V}_p(\mathbf{e_p}(t)) < 0$, if $\lambda_{min}(\Lambda_p)||\mathbf{e_p}||^2 > 2(\Gamma_p + ||\mathbf{s_p}||||\Theta_p||)$, implying that the error is bounded by

$$||\mathbf{e_p}||^* = \frac{||\mathbf{B_p^T P_p}||||\Theta_p|| + \sqrt{||\mathbf{B_p^T P_p}||^2||\Theta_p||^2 + 2\lambda_{min}(\Lambda_p)\Gamma_p}}{\lambda_{min}(\Lambda_p)}. \tag{A35}$$

By substituting the error bound (A35) in (A34) and using the relationship in (A33), we can infer that $\dot{V}_p < 0$, if $V_p > \Omega_{p1}$, where

$$\Omega_{p1} \triangleq \Gamma_p + ||\mathbf{B_p^T P_p}||||\Theta_p||||\mathbf{e_p}||^*, \tag{A36}$$

implying that,

$$V_p(\mathbf{e_p}(t)) \leq max\{V_p(\mathbf{e_p}(0)), \Omega_{p1}\}. \tag{A37}$$

**Case 2:** $\hat{k}_{\mathbf{p}} > \gamma_{\mathbf{p}}, \mathbf{s_p^T \dot{s}_p} < 0, ||\mathbf{s_p}|| \geq \varpi_{\mathbf{p}}$

Substituting the appropriate values of $\Delta\mathbf{u_p}, \dot{\hat{k}}_p$ from (6) and (15) in (A31), we have

$$\dot{V}_p \leq -\frac{1}{2}\lambda_{min}(\Lambda_p)||\mathbf{e_p}||^2 + \Gamma_p + ||\mathbf{s_p}||||\Theta_p|| + \left(-\alpha_{\mathbf{p}}\hat{k}_{\mathbf{p}}\frac{\mathbf{s_p^T s_p}}{||\mathbf{s_p}||} + k_{\mathbf{p}}||\mathbf{s_p}||\right) - (\hat{k}_{\mathbf{p}} - k_{\mathbf{p}})||\mathbf{s_p}||$$

$$\leq -\frac{1}{2}\lambda_{min}(\Lambda_p)||\mathbf{e_p}||^2 + \Gamma_p + ||\mathbf{s_p}||(||\Theta_p|| + 2k_{\mathbf{p}} - \hat{k}_{\mathbf{p}}(\alpha_{\mathbf{p}} + 1)). \tag{A38}$$

As long as $\lambda_{min}(\Lambda_p)||\mathbf{e_p}||^2 > 2(\Gamma_p + ||\mathbf{s_p}||(||\Theta_p|| + 2k_{\mathbf{p}} - \hat{k}_{\mathbf{p}}(\alpha_{\mathbf{p}} + 1)))$ inflicting an upper bound on the error to be

$$||\mathbf{e_p}||^* = \frac{||\mathbf{B_p^T P_p}||(||\Theta_p|| + 2k_{\mathbf{p}} - \hat{k}_{\mathbf{p}}(\alpha_{\mathbf{p}} + 1))}{\lambda_{min}(\Lambda_p)}$$
$$+ \frac{\sqrt{||\mathbf{B_p^T P_p}||^2(||\Theta_p|| + 2k_{\mathbf{p}} - \hat{k}_{\mathbf{p}}(\alpha_{\mathbf{p}} + 1))^2 + 2\lambda_{min}(\Lambda_p)\Gamma_p}}{\lambda_{min}(\Lambda_p)}, \tag{A39}$$

which along with (A37) and (A38) would imply that $\dot{V}_p < 0$, if $V_p > \Omega_{p2}$, where

$$\Omega_{p2} \triangleq \frac{\Gamma_p + ||\mathbf{B_p^T P_p}||||\mathbf{e_p}||^*(||\Theta_p|| + 2k_{\mathbf{p}} - \hat{k}_{\mathbf{p}}(\alpha_{\mathbf{p}} + 1))}{\varrho_p}. \tag{A40}$$

Thus, from (A32), (A38), (A39) and (A40), we have,

$$V_p(\mathbf{e_p}(t)) \leq max\{V_p(\mathbf{e_p}(0)), \Omega_{p2}\} \tag{A41}$$

**Case 3:** $\hat{k}_{\mathbf{p}} \leq \gamma_{\mathbf{p}}, \text{ any } \mathbf{s_p^T \dot{s}_p}, ||\mathbf{s_p}|| \geq \varpi_{\mathbf{p}}$

By choosing the appropriate values for $\Delta\mathbf{u_p}, \dot{\hat{k}}_p$ from (6) and (15) in (A31), it can be observed that,

$$\dot{V}_p \leq -\frac{1}{2}\lambda_{min}(\Lambda_p)||\mathbf{e_p}||^2 + \Gamma_p + ||\mathbf{s_p}||||\Theta_p|| + (-\alpha_\mathbf{p}\hat{k}_\mathbf{p}\frac{\mathbf{s_p^T s_p}}{||\mathbf{s_p}||} + k_\mathbf{p}||\mathbf{s_p}||) + (\hat{k}_\mathbf{p} - k_\mathbf{p})\gamma_\mathbf{p}$$

$$\leq -\frac{1}{2}\lambda_{min}(\Lambda_p)||\mathbf{e_p}||^2 + \Gamma_p + ||\mathbf{s_p}||(||\Theta_p|| - \alpha_\mathbf{p}\hat{k}_\mathbf{p} + k_\mathbf{p}) + \gamma_\mathbf{p}^2. \tag{A42}$$

The simplification of the last term in (A42) comes from the relationship $(\hat{k}_\mathbf{p} - k_\mathbf{p})\gamma_\mathbf{p} \leq \gamma_\mathbf{p}^2$. Hence, $\dot{V}_p < 0$, if $\lambda_{min}(\Lambda_p)||\mathbf{e_p}||^2 > 2(\Gamma_p + ||\mathbf{s_p}||(||\Theta_p|| - \alpha_\mathbf{p}\hat{k}_\mathbf{p} + k_\mathbf{p}) + \gamma_\mathbf{p}^2)$. So, the error bound for achieving the UUB property of the system is given by,

$$||\mathbf{e_p}||^* = \frac{||\mathbf{B_p^T P_p}||(||\Theta_p|| - \alpha_\mathbf{p}\hat{k}_\mathbf{p} + k_\mathbf{p})}{\lambda_{min}(\Lambda_p)}$$

$$+ \frac{\sqrt{||\mathbf{B_p^T P_p}||^2(||\Theta_p|| - \alpha_\mathbf{p}\hat{k}_\mathbf{p} + k_\mathbf{p})^2 + 2\lambda_{min}(\Lambda_p)(\Gamma_p + \gamma_\mathbf{p}^2)}}{\lambda_{min}(\Lambda_p)} \tag{A43}$$

Hence, it can be observed from (A32), (A42) and (A43) that $\dot{V}_p < 0$, if $V_p > \Omega_{p3}$ and $V_p$ is bounded as,

$$V_p(\mathbf{e_p}(t)) \leq max\{V_p(\mathbf{e_p}(0)), \Omega_{p3}\}, \tag{A44}$$

$$\Omega_{p3} \triangleq \frac{\Gamma_p + \gamma_\mathbf{p}^2 + ||\mathbf{B_p^T P_p}||||\mathbf{e_p}||^*(||\Theta_p|| - \alpha_\mathbf{p}\hat{k}_\mathbf{p} + k_\mathbf{p})}{\varrho_p} \tag{A45}$$

**Case 4:** $\hat{k}_\mathbf{p} > \gamma_\mathbf{p}$, $\mathbf{s_p^T \dot{s}_p} > 0$, $||\mathbf{s_p}|| < \varpi_\mathbf{p}$
$\dot{V}_p$ in this case can be derived from (7), (15) and (A31) as,

$$\dot{V}_p \leq -\frac{1}{2}\lambda_{min}(\Lambda_p)||\mathbf{e_p}||^2 + \Gamma_p + ||\mathbf{s_p}||||\Theta_p|| + (-\alpha_\mathbf{p}\hat{k}_\mathbf{p}\frac{\mathbf{s_p^T s_p}}{\varpi_\mathbf{p}} + k_\mathbf{p}||\mathbf{s_p}||) + (\hat{k}_\mathbf{p} - k_\mathbf{p})||\mathbf{s_p}||$$

$$\leq -\frac{1}{2}\lambda_{min}(\Lambda_p)||\mathbf{e_p}||^2 + \Gamma_p + ||\mathbf{s_p}||(||\Theta_p|| + \hat{k}_\mathbf{p}) + ||\mathbf{s_p}||^2(-\frac{\alpha_\mathbf{p}\hat{k}_\mathbf{p}}{\varpi_\mathbf{p}}) \tag{A46}$$

The condition $\lambda_{min}(\Lambda_p)||\mathbf{e_p}||^2 > 2(\Gamma_p + ||\mathbf{s_p}||(||\Theta_p|| + \hat{k}_\mathbf{p}) + ||\mathbf{s_p}||^2(-\frac{\alpha_\mathbf{p}\hat{k}_\mathbf{p}}{\varpi_\mathbf{p}}))$ needs to be satisfied to achieve $\dot{V}_P < 0$. When $||\mathbf{s_p}|| = \frac{(\hat{k}_\mathbf{p} + ||\Theta_p||)\varpi_\mathbf{p}}{2\alpha_\mathbf{p}\hat{k}_p}$, the sum of the terms in (A46) with $||\mathbf{s_p}||$ attain its maximum value of $\frac{(\hat{k}_\mathbf{p} + ||\Theta_p||)^2\varpi_\mathbf{p}}{4\alpha_\mathbf{p}\hat{k}_p}$, which gives result to the necessary upper bound of the error as,

$$||\mathbf{e_p}||^* = \sqrt{\frac{4\alpha_\mathbf{p}\hat{k}_\mathbf{p}\Gamma_p + (\hat{k}_\mathbf{p} + ||\Theta_p||)^2\varpi_\mathbf{p}}{2\alpha_\mathbf{p}\hat{k}_\mathbf{p}}} \tag{A47}$$

From (A31), (A32) and (A47), it is proven that $\dot{V}_p < 0$, if $V_p > \Omega_{p5}$, which bounds $V_p$ by,

$$V_p(\mathbf{e_p}(t)) \leq max\{V_p(\mathbf{e_p}(0)), \Omega_{p4}\}, \tag{A48}$$

$$\Omega_{p4} \triangleq \frac{\varpi_\mathbf{p}\Gamma_p + \varpi_\mathbf{p}||\mathbf{B_p^T P_p}||||\mathbf{e_p}||^*(||\Theta_p|| + k_\mathbf{p}) - ||\mathbf{B_p^T P_p}||^2||\mathbf{e_p}||^{*2}(\alpha_\mathbf{p}\hat{k}_\mathbf{p})}{\varrho_p\varpi_\mathbf{p}} \tag{A49}$$

**Case 5:** $\hat{k}_\mathbf{p} > \gamma_\mathbf{p}$, $\mathbf{s_p^T \dot{s}_p} \leq 0$, $||\mathbf{s_p}|| < \varpi_\mathbf{p}$
Selecting the corresponding values for $\Delta\mathbf{u_p}$, $\dot{k}_\mathbf{p}$ from (6) and (15) respectively, we have

$$\dot{V}_p \le -\frac{1}{2}\lambda_{min}(\Lambda_p)||\mathbf{e_p}||^2 + \Gamma_p + ||\mathbf{s_p}||||\Theta_p|| + (-\alpha_\mathbf{p}\hat{k}_\mathbf{p}\frac{\mathbf{s_p^T s_p}}{\varpi_\mathbf{p}} + k_\mathbf{p}||\mathbf{s_p}||) - (\hat{k}_\mathbf{p} - k_\mathbf{p})||\mathbf{s_p}||$$

$$\le -\frac{1}{2}\lambda_{min}(\Lambda_p)||\mathbf{e_p}||^2 + \Gamma_p + ||\mathbf{s_p}||(||\Theta_p|| + 2k_\mathbf{p} - \hat{k}_\mathbf{p}) + ||\mathbf{s_p}||^2(-\frac{\alpha_\mathbf{p}\hat{k}_\mathbf{p}}{\varpi_\mathbf{p}}) \tag{A50}$$

The condition $\lambda_{min}(\Lambda_p)||\mathbf{e_p}||^2 > 2(\Gamma_p + ||\mathbf{s_p}||(||\Theta_p|| + 2k_\mathbf{p} - \hat{k}_\mathbf{p}) + ||\mathbf{s_p}||^2(-\frac{\alpha_\mathbf{p}\hat{k}_\mathbf{p}}{\varpi_\mathbf{p}}))$ needs to be satisfied to achieve $\dot{V}_P < 0$. When $||\mathbf{s_p}|| = \frac{(2k_\mathbf{p} - \hat{k}_\mathbf{p} + ||\Theta_\mathbf{p}||)\varpi_\mathbf{p}}{2\alpha_\mathbf{p}\hat{k}_p}$, the sum of the terms in (A46) with $||\mathbf{s_p}||$ attain its maximum value of $\frac{(2k_\mathbf{p} - \hat{k}_\mathbf{p} + ||\Theta_\mathbf{p}||)^2\varpi_\mathbf{p}}{4\alpha_\mathbf{p}\hat{k}_\mathbf{p}}$, which gives result to the necessary upper bound of the error as,

$$||\mathbf{e_p}||^* = \sqrt{\frac{4\alpha_\mathbf{p}\hat{k}_\mathbf{p}\Gamma_p + (2k_\mathbf{p} - \hat{k}_\mathbf{p} + ||\Theta_\mathbf{p}||)^2\varpi_\mathbf{p}}{2\alpha_\mathbf{p}\hat{k}_\mathbf{p}}} \tag{A51}$$

From (A31), (A32) and (A51), it is proven that $\dot{V}_p < 0$, if $V_p > \Omega_{p5}$, which bounds $V_p$ by,

$$V_p(\mathbf{e_p}(t)) \le max\{V_p(\mathbf{e_p}(0)), \Omega_{p5}\}, \tag{A52}$$

$$\Omega_{p5} \triangleq \frac{\varpi_\mathbf{p}\Gamma_p + \varpi_\mathbf{p}||\mathbf{B_p^T P_p}||||\mathbf{e_p}||^*(||\Theta_p|| + 2k_\mathbf{p} - \hat{k}_\mathbf{p} - \alpha_\mathbf{p}\hat{k}_\mathbf{p}||\mathbf{B_p^T P_p}||||\mathbf{e_p}||^*)}{\varrho_p\varpi_\mathbf{p}} \tag{A53}$$

**Case 6:** $\hat{k}_\mathbf{p} \le \gamma_\mathbf{p}$, *any* $\mathbf{s_p^T}\dot{\mathbf{s}}_\mathbf{p}$, $||\mathbf{s_p}|| < \varpi_\mathbf{p}$

By choosing the appropriate values for $\Delta\mathbf{u_p}, \dot{\hat{k}}_p$ from (6) and (15) in (A31), it can be observed that,

$$\dot{V}_p \le -\frac{1}{2}\lambda_{min}(\Lambda_p)||\mathbf{e_p}||^2 + \Gamma_p + ||\mathbf{s_p}||||\Theta_p|| + (-\alpha_\mathbf{p}\hat{k}_\mathbf{p}\frac{\mathbf{s_p^T s_p}}{\varpi_\mathbf{p}} + k_\mathbf{p}||\mathbf{s_p}||) + (\hat{k}_\mathbf{p} - k_\mathbf{p})\gamma_\mathbf{p}$$

$$\le -\frac{1}{2}\lambda_{min}(\Lambda_p)||\mathbf{e_p}||^2 + \Gamma_p + ||\mathbf{s_p}||(||\Theta_p|| + k_\mathbf{p}) + \gamma_\mathbf{p}^2 + ||\mathbf{s_p}||^2(-\frac{\alpha_\mathbf{p}\hat{k}_\mathbf{p}}{\varpi_\mathbf{p}}). \tag{A54}$$

The simplification of the last term in (A54) comes from the relationship $(\hat{k}_\mathbf{p} - k_\mathbf{p})\gamma_\mathbf{p} \le \gamma_\mathbf{p}^2$. Hence, $\dot{V}_p < 0$, if $\lambda_{min}(\Lambda_p)||\mathbf{e_p}||^2 > 2(\Gamma_p + ||\mathbf{s_p}||(||\Theta_p|| + k_\mathbf{p}) + \gamma_\mathbf{p}^2 - ||\mathbf{s_p}||^2(\frac{\alpha_\mathbf{p}\hat{k}_\mathbf{p}}{\varpi_\mathbf{p}}))$. When $||\mathbf{s_p}|| = \frac{(k_\mathbf{p} + ||\Theta_\mathbf{p}||)\varpi_\mathbf{p}}{2\alpha_\mathbf{p}\hat{k}_p}$, the sum of the terms in (A46) with $||\mathbf{s_p}||$ attain its maximum value of $\frac{(k_\mathbf{p} + ||\Theta_\mathbf{p}||)^2\varpi_\mathbf{p}}{4\alpha_\mathbf{p}\hat{k}_\mathbf{p}}$, which gives result to the necessary upper bound of the error as,

$$||\mathbf{e_p}||^* = \sqrt{\frac{4\alpha_\mathbf{p}\hat{k}_\mathbf{p}(\Gamma_p + \gamma_\mathbf{p}^2) + (k_\mathbf{p} + ||\Theta_\mathbf{p}||)^2\varpi_\mathbf{p}}{2\alpha_\mathbf{p}\hat{k}_\mathbf{p}}} \tag{A55}$$

From (A31), (A32) and (A55), it is proven that $\dot{V}_p < 0$, if $V_p > \Omega_{p5}$, which bounds $V_p$ by,

$$V_p(\mathbf{e_p}(t)) \le max\{V_p(\mathbf{e_p}(0)), \Omega_{p6}\}, \tag{A56}$$

$$\Omega_{p6} \triangleq \frac{\varpi_\mathbf{p}(\Gamma_p + \gamma_\mathbf{p}^2)\varpi_\mathbf{p}||\mathbf{B_p^T P_p}||||\mathbf{e_p}||^*(||\Theta_p|| + k_\mathbf{p} - ||\mathbf{B_p^T P_p}||||\mathbf{e_p}||^*(\alpha_\mathbf{p}\hat{k}_\mathbf{p}))}{\varrho_p\varpi_\mathbf{p}} \tag{A57}$$

From (A37), (A41), (A44), (A48), (A52) and (A56), it is observed that for any possible scenario,

$$V_p(\mathbf{e_p}(t)) \leq max\{V_p(\mathbf{e_p}(0)), \underset{i}{max}\{\Omega_{pi}\}\}, \qquad i \in \{1 - 6\} \qquad \text{(A58)}$$

By following a similar procedure from (A12) to (A58) using the time-derivative of $V_p$, continuing from (A26), we can obtain the relationship,

$$V_q(\mathbf{e_q}(t)) \leq max\{V_q(\mathbf{e_q}(0)), \underset{i}{max}\{\Omega_{qi}\}\}, \qquad i \in \{1 - 6\} \qquad \text{(A59)}$$

where $\Omega_{qi}$, $i \in \{1 - 6\}$ are scalar terms similar to those in (A36), (A40), (A45), (A49), (A53) and (A57). Hence, from (A58), (A59) the overall Lyapunov function, $V$ in (A5) is ultimately bounded by,

$$V(\mathbf{e_p}(t), \mathbf{e_q}(t)) \leq max\{V_q(\mathbf{e_p}(0), \mathbf{e_q}(0)), \Omega\} \qquad \text{(A60)}$$

where $\Omega = (\underset{i}{max}\{\Omega_{pi}\} + \underset{j}{max}\{\Omega_{qj}\})$, $i, j \in \{1 - 6\}$, and the closed-loop system remains UUB.

It can be noted that unlike the predictive approach in [49,50], the proposed controller does not mandate any constraint on the derivative of the time-varying delay. Further, the knowledge of the upper bound of the delay is used only in the parameter selection and hence, the knowledge of the instantaneous value of the delay and its time-derivatives is not required in the controller design. Similarly, the knowledge of the overall parametric uncertainties $\sigma_\mathbf{p}, \sigma_\mathbf{q}$ are used neither in the control laws nor in the adaptive laws. Hence, despite the boundedness of the uncertainties, the knowledge of their bounds are not needed by the controller. □

**Remark A1.** *The bounds $\Omega_{pi}$, $\Omega_{qj}$, $\forall i, j \in \{1 - 6\}$ are dependant on the gain parameters $\alpha_\mathbf{p}, \alpha_\mathbf{q}$ and the delay in the system h. The performance maximizes with higher gains in the presence of minimal delays. However, high gains lead to large control inputs, which is undesirable in the practical scenario.*

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
