# Peer review of "Adaptive Robust Control for Quadrotors with Unknown Time-Varying Delays and Uncertainties in Dynamics"

_drones, doi:10.3390/drones6090220_

Round 1

Reviewer 1 Report

This paper presents an adaptive control law for quadcopters subject to external disturbances and delays. This work could be of some interest, but the authors need to address several key issues:

- What is the need for accounting for time delays on quadcopters? Current single-board computers for these vehicles are so fast that they can easily handle complex controllers with substantially no time delays.

- This work lacks proofs, and it is impossible to check whether the proposed controllers are correct or not.

- The proposed control laws have switching mechanisms. It is unclear how the proposed controller responds to chattering from a theoretical standpoint. How can \alpha avoid chattering? Are the Authors trying to create a boundary layer through \hat{\omega}? Also, I may be missing something but the first and second equations on the right-hand sides of (12) and (24) seem identical. Finally, additional comments on the potential chattering problem and how it is addressed should be provided.

- The use of control laws that seem to employ a variable structure approach makes me wonder how the upper bounds on the uncertainties are not employed. Having presented proofs would have helped clarifying this doubt.

Additional concerns are the following:

- It is striking that neither of the two control laws has adaptive rates to regulate convergence. A great feature of any adaptive controller is some positive(-definite) tunable element that allows steering the rate of the convergence of the underlying Lyapunov function.

- The authors assume that \tau_p \neq 0 at all times. This prevents the UAV from performing aggressive maneuvers, such as free falls.  What restrictions prevent this from happening?

- Assumption 1 includes the assumption on the boundedness of constants, such as \Delta m (for example), which are clearly bounded. These assumptions can be removed. Also, how can a scalar (\Vert \Delta J \Vert) be bounded by a matrix (\mathbf{I})?

- What do the Authors mean by "feasible" in Assumption 2?

- The relation between e_p and e_{1p} should be presented explicitly.

- The Authors should provide additional discussion on the kind of UAV they consider. Is it an actual one? What size? What is the mass without the payload? How can the mass of the UAV be constant (1.9kg) and its payload vary in multiple scenarios (0.3kg and 0.7kg)?

- In Section 4, the characterization of the external disturbances is unclear. 

- Why is the UAV tasked with keeping a constant yaw angle, instead of tasking it to point toward a target or tangential to the UAV?

- The authors should add the following relevant works to the list of papers cited and discuss the relative relevance of their work:

+https://hal.archives-ouvertes.fr/hal-03561999/document

+https://ieeexplore.ieee.org/document/9138676

+ https://www.mdpi.com/1424-8220/21/7/2401

Reviewer 2 Report

·      Introduction is not concise and takes too long to describe what the authors are explicitly trying to do.

o   Paper would benefit from stating contributions earlier and tightening up introduction. For example, the second paragraph mentions “a priori” fives times

o   There are a few grammatical issues within the introduction as well.

o   Some of the introduction could be moved to a literature review section.

·      Page 5, “the (21))” seems to be missing some words/context.

·      Generally this paper would benefit from

·      Specifically stating a target vehicles and payloads for parameters used in modeling

·      Comparing to other control techniques

·      Significant revision of the writing as there are too many English errors to list or count

·      Statement of the exact problem and proposed solution

Round 2

Reviewer 1 Report

The authors addressed all my concerns

Author Response

Thanks for your valuable comments.